# TranSynergy: Mechanism-driven interpretable deep neural network for the synergistic prediction and pathway deconvolution of drug combinations

Qiao Liu[1], Lei Xie[1,2,3,4]*

**1** Department of Computer Science, Hunter College, The City University of New York, New York, United States of America, **2** Ph.D. Program in Computer Science, The City University of New York, New York, United States of America, **3** Ph.D. Program in Biochemistry and Biology, The City University of New York, New York, United States of America, **4** Helen and Robert Appel Alzheimer's Disease Research Institute, Feil Family Brain & Mind Research Institute, Weill Cornell Medicine, Cornell University, New York, United States of America

\* lxie@iscb.org

**Data Availability Statement:** All scripts and data are available from https://github.com/qiaoliuhub/drug_combination. Other relevant results data are

## Abstract

Drug combinations have demonstrated great potential in cancer treatments. They alleviate drug resistance and improve therapeutic efficacy. The fast-growing number of anti-cancer drugs has caused the experimental investigation of all drug combinations to become costly and time-consuming. Computational techniques can improve the efficiency of drug combination screening. Despite recent advances in applying machine learning to synergistic drug combination prediction, several challenges remain. First, the performance of existing methods is suboptimal. There is still much space for improvement. Second, biological knowledge has not been fully incorporated into the model. Finally, many models are lack interpretability, limiting their clinical applications. To address these challenges, we have developed a knowledge-enabled and self-attention transformer boosted deep learning model, TranSynergy, which improves the performance and interpretability of synergistic drug combination prediction. TranSynergy is designed so that the cellular effect of drug actions can be explicitly modeled through cell-line gene dependency, gene-gene interaction, and genome-wide drug-target interaction. A novel Shapley Additive Gene Set Enrichment Analysis (SA-GSEA) method has been developed to deconvolute genes that contribute to the synergistic drug combination and improve model interpretability. Extensive benchmark studies demonstrate that TranSynergy outperforms the state-of-the-art method, suggesting the potential of mechanism-driven machine learning. Novel pathways that are associated with the synergistic combinations are revealed and supported by experimental evidences. They may provide new insights into identifying biomarkers for precision medicine and discovering new anti-cancer therapies. Several new synergistic drug combinations have been predicted with high confidence for ovarian cancer which has few treatment options. The code is available at https://github.com/qiaoliuhub/drug_combination.

within the manuscript and its Supporting Information files.

**Funding:** This work was supported by Grant Number R01GM122845 from the National Institute of General Medical Sciences (NIGMS, to LX) and Grant Number R01AD057555 of National Institute on Aging of the National Institute of Health (NIH, to LX and QL) as well as CUNY High Performance Computing Center (to LX). The funders had no role in study design, data collection and analysis, decision to publish, or preparation of the manuscript.

**Competing interests:** The authors have declared that no competing interests exist.

## Author summary

The number of anti-cancer drugs has been consistently and quickly growing. They are mainly used as standardized mono-therapy. Drug combinations show substantial advantages over the anti-cancer mono-therapy. Cancer cells treated with the mono-therapy could later activate bypassing pathways and harbor drug resistances. Drug combinations can alleviate this issue by using a smaller doses of each anti-cancer drug or targeting multiple oncogenic pathways. However, the investigation of all anti-cancer drug combinations using experimental methods is costly and time-consuming. Machine learning provides an attractive solution to screening synergistic drug combinations, but it is a black-box and not easy to explain. We have developed a knowledge-enabled deep learning model, TranSynergy, to predict synergistic drug combinations and have demonstrated that our model outperformed other state-of-the-art methods. A novel Shapley Additive Gene Set Enrichment Analysis (SA-GSEA) method is introduced to improve the interpretability of the machine learning model. Using TransSynergy and SA-GSEA, we can deconvolute genes responsible for the synergistic drug combination, suggesting the potential of machine learning in developing precision anti-cancer therapy.

## Introduction

With the advance of understandings of cancer cell disorders, more and more anti-cancer drugs have been designed and are under investigation. However, the "one drug, one target" drug monotherapy suffers limited efficiency due to inherent or acquired resistance [1–3]. Drug combination therapy is a more effective strategy to solve this challenging problem [4–8]. In addition to cancer, synergistic drug combinations have several successful applications in the treatment of other diseases, such as AIDS [9,10], and fungal or bacterial infections [11–13]. Thus, the selection of efficient drug combination therapy for pathogens emerges as a compelling treatment strategy. Considering that the number of anti-cancer drugs has increased drastically, the possible combination of all these drugs has also become enormous [14,15]. The existing experimental method requires a large number of samples with different drug doses and cancer cells [16], thus it is infeasible to exhaust all the possible drug combinations. The computational method can be used to pre-select drug combinations with high synergy at lower cost and with more efficiency. The recent advancement of computational modeling, especially the deep learning technique, has dramatically increased the prediction power of computational models and has many promising applications in the biomedical field. The combination of computational and experimental methods can improve the effectiveness of the drug combination discovery.

The deep learning model has been shown to have superior performance to conventional machine learning algorithms in many biomedical applications [17,18]. High-quality experimental drug combination datasets are necessary for the success of deep learning. With the advancement of high throughput drug combination screening tests, the number of samples grows fast so that the data size limitation is considerably alleviated [19–23]. DeepSynergy is a state-of-the-art deep learning-based prediction model for the prediction of the synergistic drug combination. It has been trained using the dataset released by Merck [24]. In addition to suboptimal performance, the issue for this model is that the interpretation of the model is limited by the way adopted to represent drugs and cell lines as well as the model architecture. For instance, it is not easy to associate the contribution or feature importance of drug descriptors,

including toxophores, physicochemical properties, and fingerprints, with the mechanism of drug action in cells using a feedforward neural network [25–27].

Recent studies have shown that gene-gene interaction and drug mechanism of action drew more and more attention in the synergistic drug combination study [28]. In addition, cell line drug sensitivity strongly depends on whether the drug directly or indirectly inhibits the essential gene of the cell line [22]. Thus, it is desirable to incorporate information from the gene-gene interaction network, gene dependency, and drug-target association into the computational model. To this end, we implemented a mechanism-driven and self-attention boosted deep learning model TranSynergy for the prediction of synergistic drug combinations and the deconvolution of cellular mechanisms contributing to them. We applied the random walk with restart algorithm (RWR) on a protein-protein interaction (PPI) network to infer a novel drug-target profile as the drug features. For the features of each cell line, we used gene expressions or gene dependencies profile. These mechanism related features make the model readily interpretable. Furthermore, we applied the self-attention transformer to encode the gene-gene interactions responsible for the synergistic drug combination. Attention mechanisms have been widely used in image processing and natural language processing [29–31] and has shown promise in the predictive modeling of nucleic acid sequences [32]. When combining the network propagated drug target profile, gene dependency, and gene expression, TranSynergy outperforms the state-of-the-art model. To reveal novel genes that are associated with the synergistic drug combination from the learned biological relations in the TranSynergy model, we developed a novel Shapley Additive Gene Set Enrichment Analysis (SA-GSEA) based on SHAP [33]. The revealed novel gene set may serve as a patient-specific biomarker for precision medicine or drug targets for discovering new cancer combination therapy. We further applied the model for the prediction of novel synergistic drug combinations targeting cancers that have few treatment options. Given the emergence of next-generation sequencing technology, the transcriptome of patient-derived cancer cells can be readily obtained [34]. The TranSynergy can be used to predict and interpret the synergistic drug combination in distinct patient-derived cancer cells. Our study shows the potential of a mechanism-driven interpretable machine learning model in the application of personalized cancer treatment.

## Results

### TranSynergy architecture

TranSynergy is a transformer boosted deep learning model for the prediction of drug combination synergy. It includes three major components, input dimension reduction component, self-attention transformer component, and output fully connected component (Fig 1). The input features are composed of three vectors. Each vector has 2401 dimensions, forming a 2401x3 matrix. The first two columns are the representations of two drugs. The third column is the representation of the cell lines. The drug feature is a drug-target interaction profile on 2401 selected genes. The cell line feature includes the gene expression or gene dependency of 2401 genes. It is worth mentioning that the number of columns changes to 4 when both gene expression and dependency are used for the representation of cells. In the matrix, each row corresponds to a gene or protein, and encodes the impact of drug on the gene. The input dimension reduction component is a single-layer neural network to reduce the dimension of input. The modified transformer component takes the output from the first component and includes a scaled dot product based self-attention mechanism module. Here, the self-attention is applied to model gene-gene interactions. It is also worth noting that we customized the transformer model by removing the positional encoding layer since the order input feature dimensions should be irrelevant to the final prediction. Then the final output of the predicted

Dimension reduction                Transformer                Fully Connected

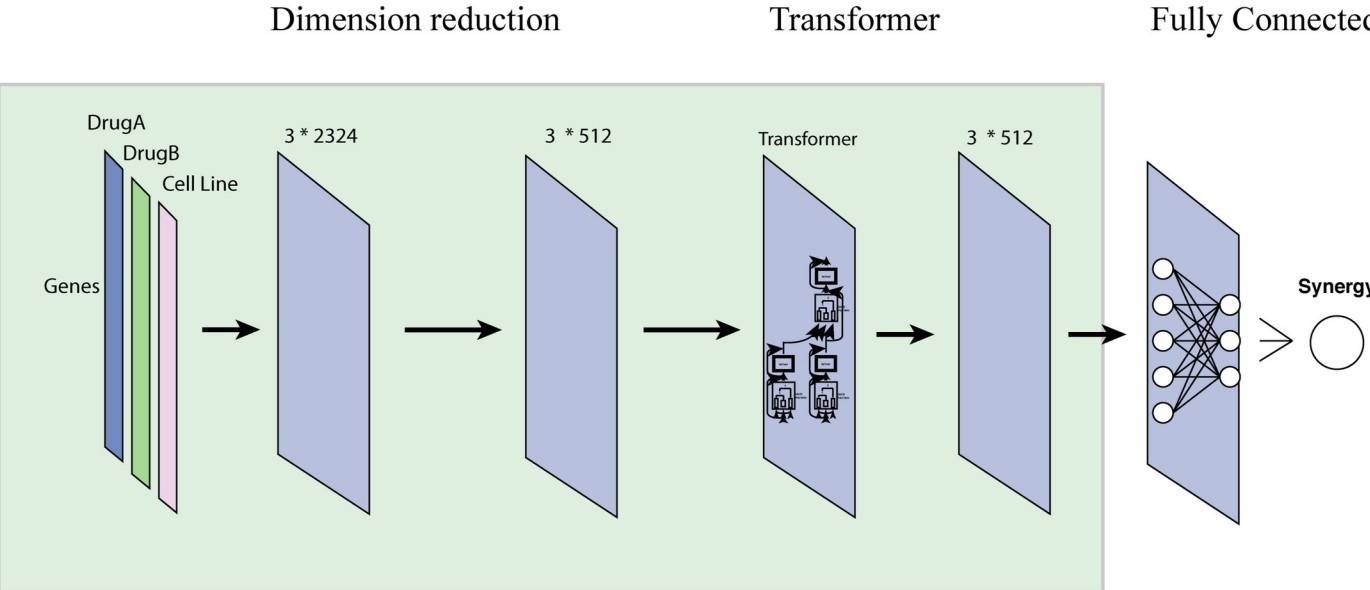

**Fig 1. The architecture of TranSynergy.** The input features include vector representations of Drug A, Drug B, and cell line vector, respectively. The first input dimension reduction component reduces the input dimension from 2401x3 to 512x3. The second component is a scaled dot product self-attention transformer. The third component is a fully connected neural network. Be noted that input matrix dimension changes to 2401x4 when both gene expression and gene dependency profiles are used for cell line representation.

synergy score comes from a fully connected neural network. The hyperparameters used in each component and training process are listed in S1 Table.

The input of our deep learning model includes the vector representations of two drug molecules in the drug combination and a cell line that is treated by the drug combination. One popular strategy is to use the physicochemical properties, toxicophores, or extended connected fingerprints (ECFP) that are derived from the molecular structure of chemical compounds for the representation of drugs [24]. The disadvantage of using the chemical structure as the feature is that it is not straightforward to establish a relationship between the physicochemical properties of drugs and the cellular mechanism of drug action. Biological representation based on drug-target interaction profile is an alternative strategy to infer the drug representation vector [35]. Drug target information that is collected from databases, including DrugBank and ChEMBL [36,37], are mainly proteins that can directly associate with drugs. We also need to encode the effect of drugs on down-stream non-target proteins and the whole biological system. The protein-protein interaction network is utilized to infer the drug response of the non-target proteins considering that the protein-protein interaction mediates information transmission in the biological system. We apply the RWR algorithm to simulate this network propagation process (Fig 2). Compared with the chemical information-based approaches for drug representation, the target-based representation of drug molecules has the following advantages. Firstly, drug target information is closely related to the cellular response to the drug treatment at both the molecular level and system level. Secondly, it makes it possible to explain the model output, drug combination synergy, in terms of the contribution of each protein or gene.

Because drug-combination therapy has cell line-specific responses, another important component of inputs is the cell line vector representation. The DeepSynergy uses gene expression profile as the cell line vector representation [24]. We have applied a novel alternative strategy to infer cell line vector representation. The gene essentiality varies in the different cell lines

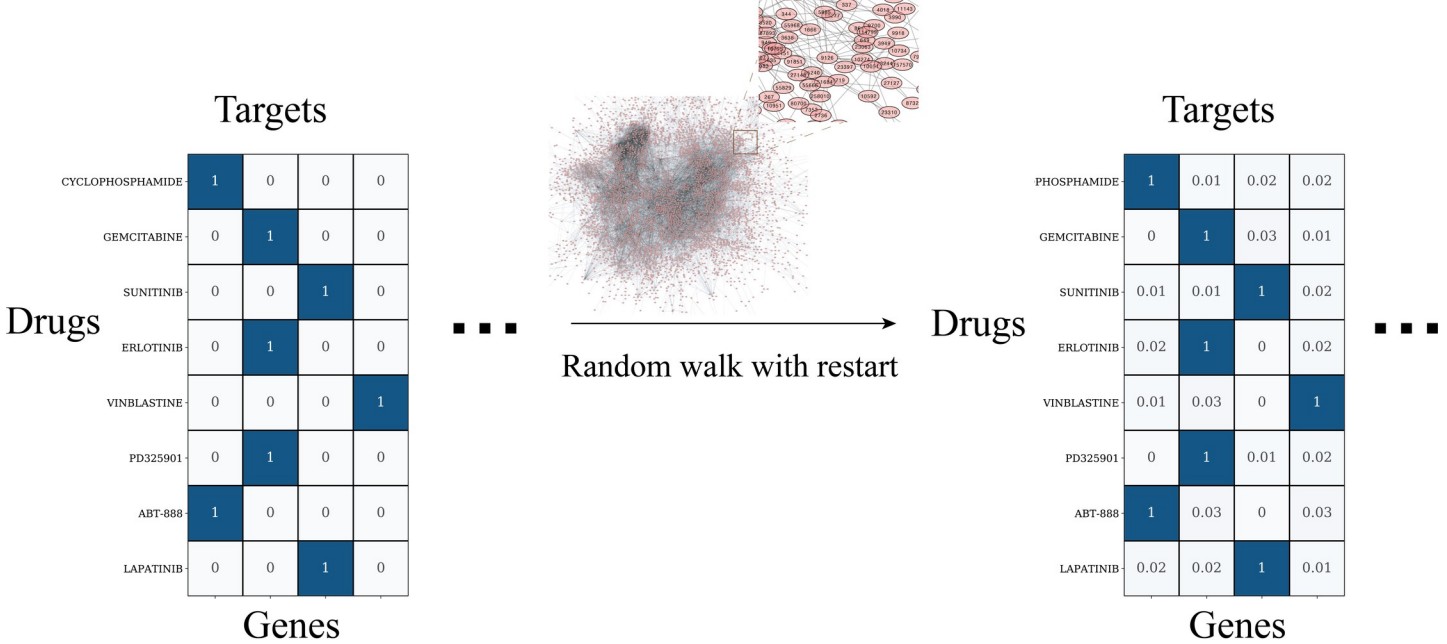

**Fig 2. Illustration of genome-wide drug-target profile.** Observed drug target profile is processed with RWR to infer drug effects on both targets and non-target proteins.

and plays a critical role in anti-cancer drug sensitivity. Intuitively, drugs that affect essential proteins will cause the cell to have a more devastating response. Thus, we take gene essentiality or gene dependence into account and this information is one of the inputs into our model. This information has been collected by the BROAD institution using experimental methods [38]. They have performed a genome-wide loss-of-function screening with pooled RNAi or CRISPR library and have investigated the resulted cellular response.

## TranSynergy outperforms state-of-the-art model and shows superior performance across different tissues

To evaluate the TranSynergy architecture on drug combination synergy score prediction, we tested our model in three different scenarios and compared it with the state-of-the-art model DeepSynergy.

We first assessed the model performance in the setting of leave drug combination out (Fig 3A). We performed a nested five-fold cross-validation using the same data for both models. We split the data into five folds based on drug combinations such that the drug combination in one fold would not appear in another fold. Four folds of data were used in the training and hyperparameter tuning stage and one fold was held out to test model performance. We performed the training/testing procedure five times with each fold held out. TranSynergy significantly outperformed the DeepSynergy model for the synergy score prediction (Table 1). The mean square error (MSE) of the TranSyerngy model was 232, significantly lower than that of DeepSynergy (p-value < 0.001). The accuracy improved by 3% and 5% when measured with Pearson's correlation and Spearman's correlation, respectively. We also showed that TranSynergy presented a superior performance in classification scenarios (Table 2). PR-AUC improved by 5.6% when comparing TranSynergy with DeepSynergy, although the improvement of ROC-AUC was by a small margin. It is noted that other genomic features of cell lines

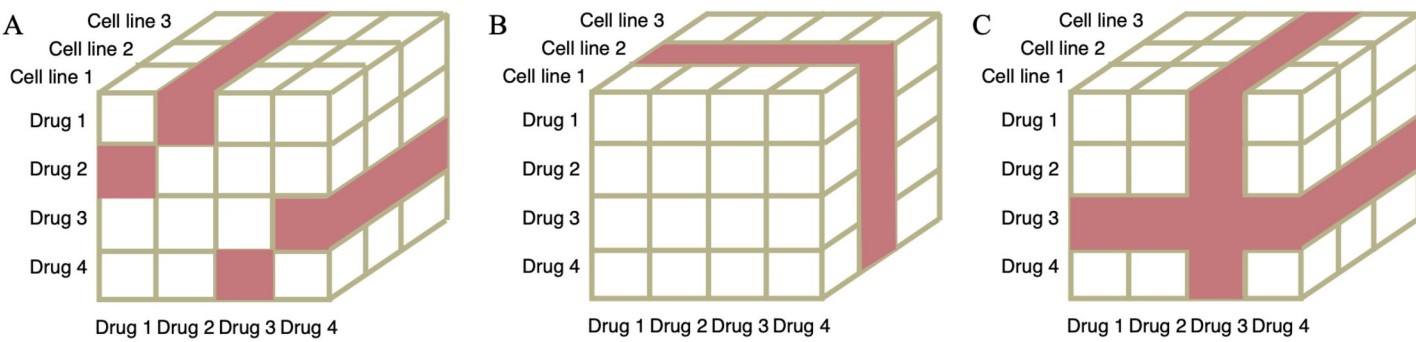

**Fig 3. Illustration of three training/test dataset splitting scenarios.** Each data point is composed of two drugs and a cell line. The data samples in white and red are the training/validation dataset and test dataset, respectively. A) Leave drug combination out scenario. In this example, drug pairs, including drug1+drug2 and drug3+drug4, are colored red and held out as test dataset. B) Leave cell line out scenario. In this example, the cell line 2 is colored red and held out as test dataset. C) Leave drug out scenario. In this example, the drug 3 and all drug pairs including the drug 3 are colored red and held out as test dataset.

(mutations and CNVs) were not used in both TranSynergy and DeepSynergy. Considering that they could provide additional information relevant to the drug mode of action, especially for targeted therapy, it would be interesting to incorporate extra genomic features into TranSynergy in the future.

We further explored the performance of the TranSynergy model on 6 different tissues: colon, breast, melanoma, ovarian, prostate, and lung. The average Pearson correlation coefficient between ground truth synergy scores and predicted scores were 0.741 for colon cancer cells, 0.742 for breast cancer cells, 0.705 for melanoma, 0.788 for ovarian cancer cells, 0.660 for prostate cancer cells, and 0.738 for lung cancer cells (Fig 4A and 4B). The Pearson correlation coefficients of the TranSynergy across cell lines ranged from 0.616 to 0.858, and the Spearman correlation coefficients ranged from 0.592 to 0.863 for (Fig 4C and 4D). Compared with DeepSynergy, the superior performance of TranSynergy was consistent across tissues except for the prostate. The prostate tissue had the lowest Pearson correlation coefficient among all tissues included in our data. VCAP, which is the only prostate cancer cell line, had the third-lowest Pearson correlation coefficient and Spearman correlation coefficient across all cell lines. The mediocre performance of the VCAP cell line could be due to that it is not closely similar to other cell lines. To reflect relationships between cell lines, we performed the t-SNE analysis to visualize the high dimensional vector representation of cell lines mapped to a 2D space with the first two t-SNE components (Fig 5). It showed that the VCAP cell line is located on the edge of one cluster, which indicates the VCAP cell line might have distinctive features. We also noticed that five cell lines were isolated from the remaining cell lines in the t-SNE plot. They included T47D and OCUBM from the breast tissue, HCT116 from the colon tissue, as well as UWB1289 and OVCAR3 from ovarian tissue. The variances of these three tissues were higher than those of the other three tissues. These larger variances among cell line features within these three tissues might contribute to the observed larger performance variances.

**Table 1. Performance comparison of TranSynergy and DeepSynergy models in regression scenarios.**

| Model | Drug features | Cell line features | MSE | Spearman Correlation | Pearson Correlation |
|---|---|---|---|---|---|
| TranSynergy | Network propagated drug target profile | Gene dependency + Gene expression | 231 ± 21 | **0.730±0.016** | **0.746±0.018** |
| DeepSynergy | Physicochemical properties, toxicophores and fingerprints | Gene expression | 243 ± 17 | 0.698±0.016 | 0.726±0.015 |
| Statistical significance (p-value) | | | <0.0001 | <0.0001 | 0.0001 |

**Table 2. Performance comparison of TranSynergy with different cell line features to DeepSynergy model in classification scenarios.**

| Models | Drug features | Cell line features | PR-AUC | ROC-AUC |
|---|---|---|---|---|
| TranSynergy | Network propagated drug target profile | Gene dependency | 0.625±0.020 | **0.908±0.011** |
| | | Gene expression | 0.618±0.016 | 0.901±0.009 |
| | | Gene expression + Gene dependency | **0.627±0.013** | 0.907±0.011 |
| DeepSynergy | Physicochemical properties, toxicophores and fingerprints | Gene expression | 0.594±0.021 | 0.900±0.009 |
| | Statistical significance* (p-value) | | <0.0001 | 0.0134 |

*TranSynergy that used gene expression+Gene dependency cell line features was used to compute the statistical significance of performance difference between TranSynergy and DeepSynergy.

We additionally tested our models in the most challenging leave cell out scenario (Fig 3B). We used an affinity propagation method to divide the cell lines into 4 clusters based on cell line features. Since 28 cell lines are in the biggest cluster, we used them as the training dataset and held out the remaining cell lines as the test dataset. In this way, the cell lines in the test dataset are the most different from those in the training dataset. All the cell lines have a similar amount of data (S4 Table). The leave cell out is a robust method to test the generalization power of the models. As shown in Tables 3 and 4, the performance of the TranSynergy is slightly better than DeepSynergy but both are lower than the first scenario.

Similar to the leave cell out scenario, we split the drugs into 5 clusters with the affinity propagation method in leave drugs out scenario (Fig 6). We chose drugs in one cluster and held out all drug pairs including those selected drugs as the test set (Fig 3C). The remaining drug pairs are the training/validation set. In this scenario, TranSynergy significantly outperformed Deep-Synergy, but both were lower than the first two scenarios (Tables 3 and 4).

## Mechanism-driven drug representations are critical for superior model performance

To investigate whether the gene-gene interaction network propagation step is essential for the model performance, we compared the TranSynergy model trained with only observed drug target information with that trained with network propagated drug target information. The observed drug target vector is a binary 2401-dimension vector that indicates whether a drug physically binds to the corresponding protein. The cell line vector representation used in this comparison is based on the cell line dependency. Models trained with the network propagated drug target vector representation showed superior performance in comparison to those with observed drug target information (Table 5). Given that the model excluding cell line features had inferior performance in comparison to other models with cell line features, we concluded that the cell line feature contributed to the synergy score prediction (Table 5). We then investigated three different representations for cell lines, gene dependency, gene expression, or gene dependency and gene expression. When we represented cell lines with gene dependency profile only or both gene dependency and gene expression profiles, the model exhibited superior performance to that representing cells with gene expression only (Table 5).

Furthermore, we implemented two novel infrastructures, TranSynergyCI and TranSynergyGNF, to investigate whether the prediction could be improved further by integrating extra chemical information. TranSynergyCI united the physiochemical, toxophores, and fingerprint information of each drug. The difference between TranSynergy and TranSynergyCI is that the latter concatenated the aforementioned features with the output of the Transformer as the input of the last fully connected component (S1 Fig). It is worth mentioning that the

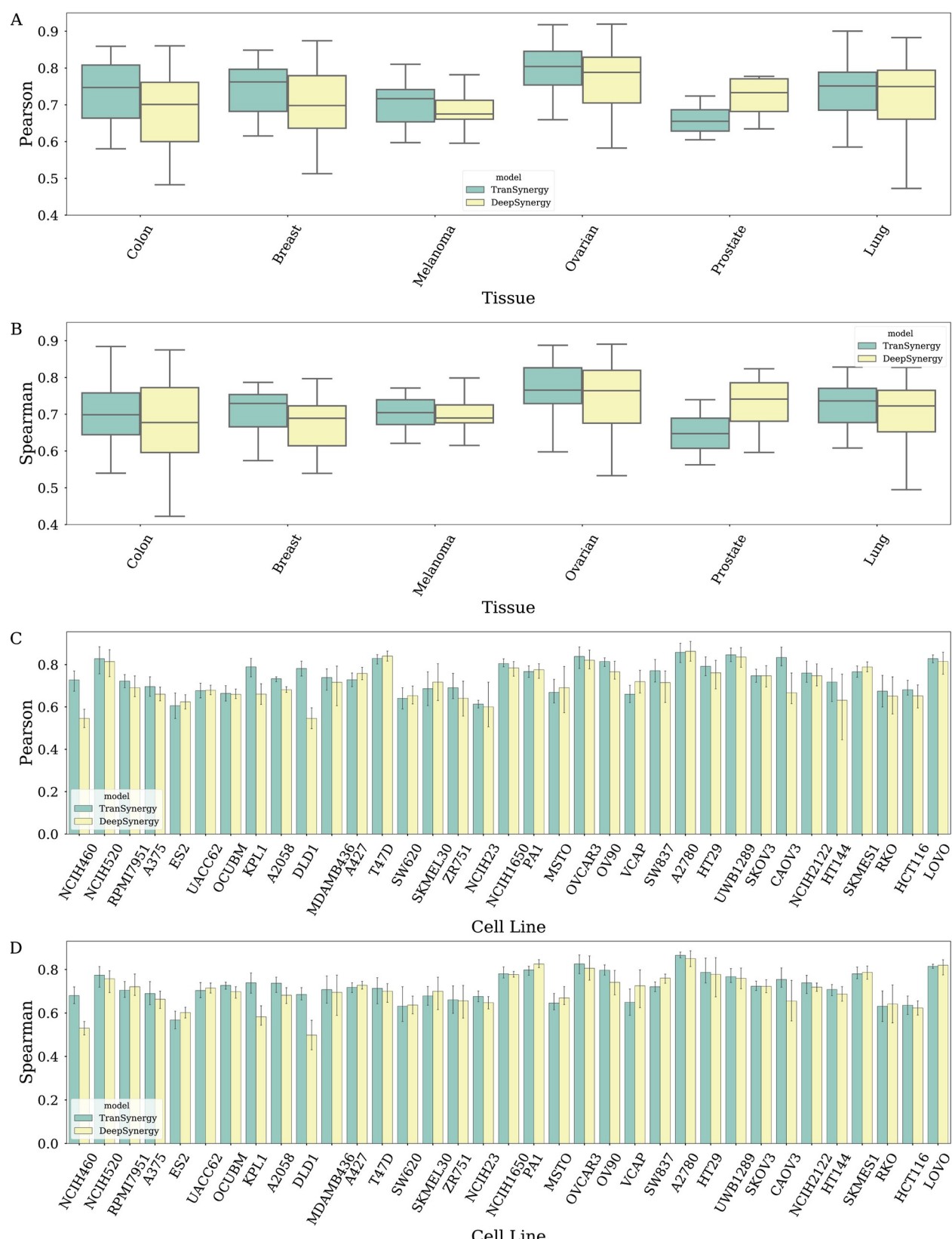

**Fig 4. Tissue-specific and cell line-specific prediction performances of TranSynergy and DeepSynergy.** The top two panels (A and B) are tissue-specific performance with Pearson correlation and Spearman correlation as the metrics. The bottom two panels (C and D) are cell line-specific performance with Pearson correlation and Spearman correlation as the metrics.

DeepSynergy used the same features for the drug representation. TranSynergyGNF also integrated chemical information to the model input, however, it applied graph convolutional neural network (GCN) to extract graphical neural fingerprints from drug chemical structures (S2 Fig). More specifically, the GCN layer took drug's atoms and bonds as inputs. Intuitively, the drug was represented as a graph structure comprised of atoms as the nodes of the graph, and bonds as the edges of the graph. The output of this GCN layer is used as the representation of each drug. The same with TranSyenrgyCI, the outputs of GCN layers are concatenated with the output of the transformer and served as the input of the last fully connected component. Both these two models showed the inferior performance to TranSynergy (S2 and S3 Tables). This illustrates adding extra chemical information may not be necessary to boost the performance any further in this experimental setup.

## Shapley additive gene set enrichment analysis reveals novel oncogenic signatures associated with synergistic drug combinations

We propose the use of Shapley Additive Gene Set Enrichment Analysis (SA-GSEA) to determine the oncogenic signature and the underlying mechanisms associated with each synergistic drug combination (see methods for details). Shapley additive value is a powerful way to characterize the feature contribution to the final prediction in each instance [33]. Because each of the features in TranSynergy corresponds to a gene, the Shapley value essentially indicates the attribute of each gene to the synergy prediction. Moreover, genes could be ranked based on the Shapley value of each input gene feature. This makes it feasible to perform GSEA analysis. As an example, we explored seven samples, which had high synergistic scores and were accurately predicted (Table 6 and S3–S9 Figs). These seven samples are from three cell lines, T47D,

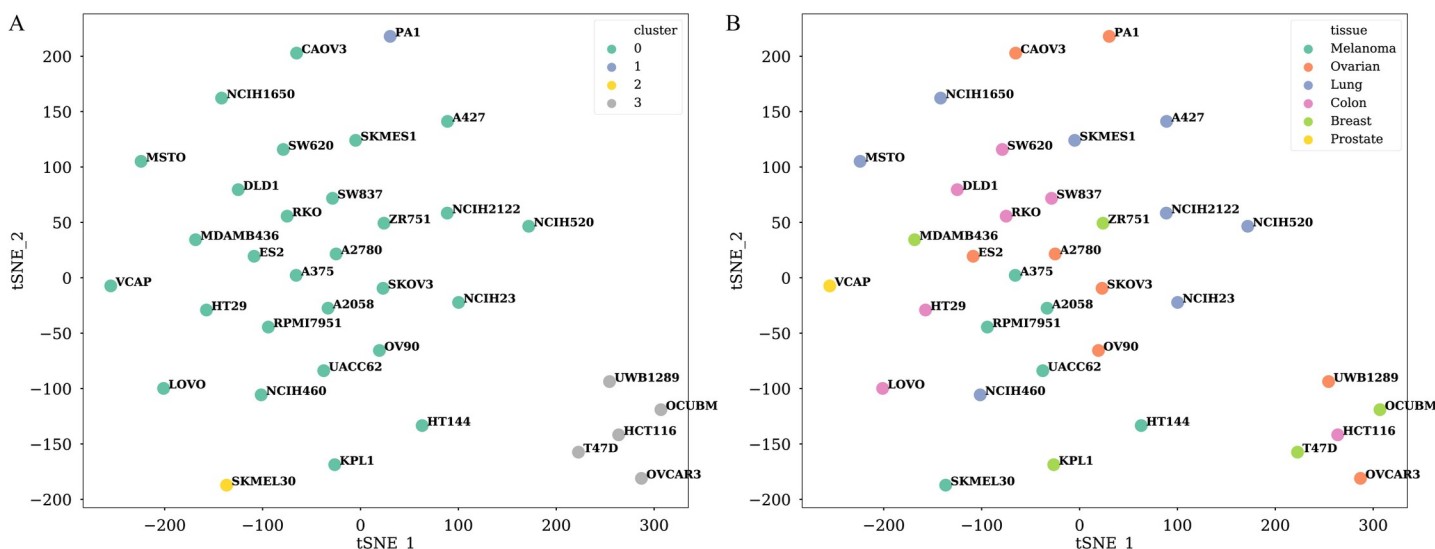

**Fig 5. Visualization of different cell lines with t-SNE analysis.** High dimensional cell line vector representations are projected into 2-D space with the first two t-SNE components. A) Different colors indicate assigned clusters of each cell line by the affinity propagation method. B) Different colors indicate different tissues of each cell line.

**Table 3. Performance of TranSynergy and DeepSyenrgy on leave cell out scenario and leave drug out scenario for the regression task.**

| Model | Leave cells out | | Leave drugs out | |
|---|---|---|---|---|
| | Spearman Correlation | Pearson Correlation | Spearman Correlation | Pearson Correlation |
| TranSynergy | **0.552 ± 0.013** | **0.513 ± 0.019** | **0.477 ± 0.040** | **0.457 ± 0.031** |
| DeepSynergy | 0.547 ± 0.010 | 0.501 ± 0.010 | 0.453 ± 0.025 | 0.437 ± 0.020 |
| Statistical significance (p-value) | 0.0716 | 0.0013 | 0.0032 | 0.0001 |

CAOV3, and MSTO-211H. For the synergistic drug pairs in T47D, BEZ-235 is one of the notable drugs. The BEZ-235 can inhibit PI3K/AKT/mTOR pathway, which is dysregulated in breast cancer [39]. In the SA-GSEA of cell line gene dependencies profile, the RAF oncogenic signature is significantly enriched. It has been established that the RAS/RAF/MEK/ERK pathway and PI3K/AKT/mTOR pathway are closely interconnected components and form feedback loops in breast cancer [40]. This may also suggest the underlying mechanism following the inhibition of the PI3K/AKT/mTOR pathway by BEZ-235 is related to the RAS/RAF/MEK/ERK pathway. The drug combination of ETOPOSIDE and MK-8669 is synergistic in the CAOV3 cancer cell line. ETOPOSIDE targets TOP2A and TOP2B, two topoisomerase components. The inhibition of them was believed to cause a DNA double-strand break [41]. MK-8669 targets mTOR, a crucial component in the PI3K/AKT/mTOR pathway. TUBB also shows surprising high importance in the SA-GSEA of MK-8669 drug targets (S3 Fig). This indicates that inhibition of mTOR or TUBB in combination with DNA damage can have a synergistic effect. In the SA-GSEA of cell line features, the JAK2 oncogenic signature is enriched in the ranked gene list based on their Shapley values. As suggested in other studies, the JAK2 oncogenic signature implies a novel pathway affected by this drug combination therapy [42]. For synergistic drug pairs in the MSTO-211H cell line, one of the drugs is DASATINIB. The DASATINIB targets BCRABL, SRC, Ephrins, and GFR. For the combination of PACLITAXEL and DASATINIB in the MSTO-211H cell line, the PACLITAXEL targets tubulin and microtubule-associated proteins. From the SA-GSEA of cell line genes, genes in the ESR1 and BMI1 oncogenic signature gene sets are top-ranked (S7 Fig) [43]. BMI1 was shown to be involved in the DNA-damage-repair process [44]. The inhibition of these pathways combining the restrain on mitosis can have high anti-tumor activities in Mesothelioma cancer [45]. For the combination of DASATINIB and ABT-888 in MSTO-211H, the SNF5 pathway is ranked on the top (S9 Fig). It is worth mentioning that both SNF5 and PARP1/2 play key roles in the DNA repair process [46,47].

## Novel drug combination prediction

Although the majority of drug combinations composed of the drugs available in this study have been explored experimentally [20], there were still 3650 novel drug pairs that had yet to be tested. For these 3650 novel samples, Table 7 lists the top 10 pairs which have higher cell line-wise z-scores (S5 Table). It is noted that ETOPOSIDE, which targets TOP2A and TOP2B,

**Table 4. Performance of TranSynergy and DeepSyenrgy on leave cell out scenario and leave drug out scenario for the classification task.**

| Model | Leave cells out | | Leave drugs out | |
|---|---|---|---|---|
| | ROC-AUC | PR-AUC | ROC-AUC | PR-AUC |
| TranSynergy | **0.808 ± 0.017** | **0.372 ± 0.022** | **0.778 ± 0.032** | **0.406 ± 0.106** |
| DeepSynergy | 0.805 ± 0.017 | 0.360 ± 0.022 | 0.753 ± 0.030 | 0.378 ± 0.075 |
| Statistical significance (p-value) | 0.3328 | 0.0908 | 0.0034 | 0.1379 |

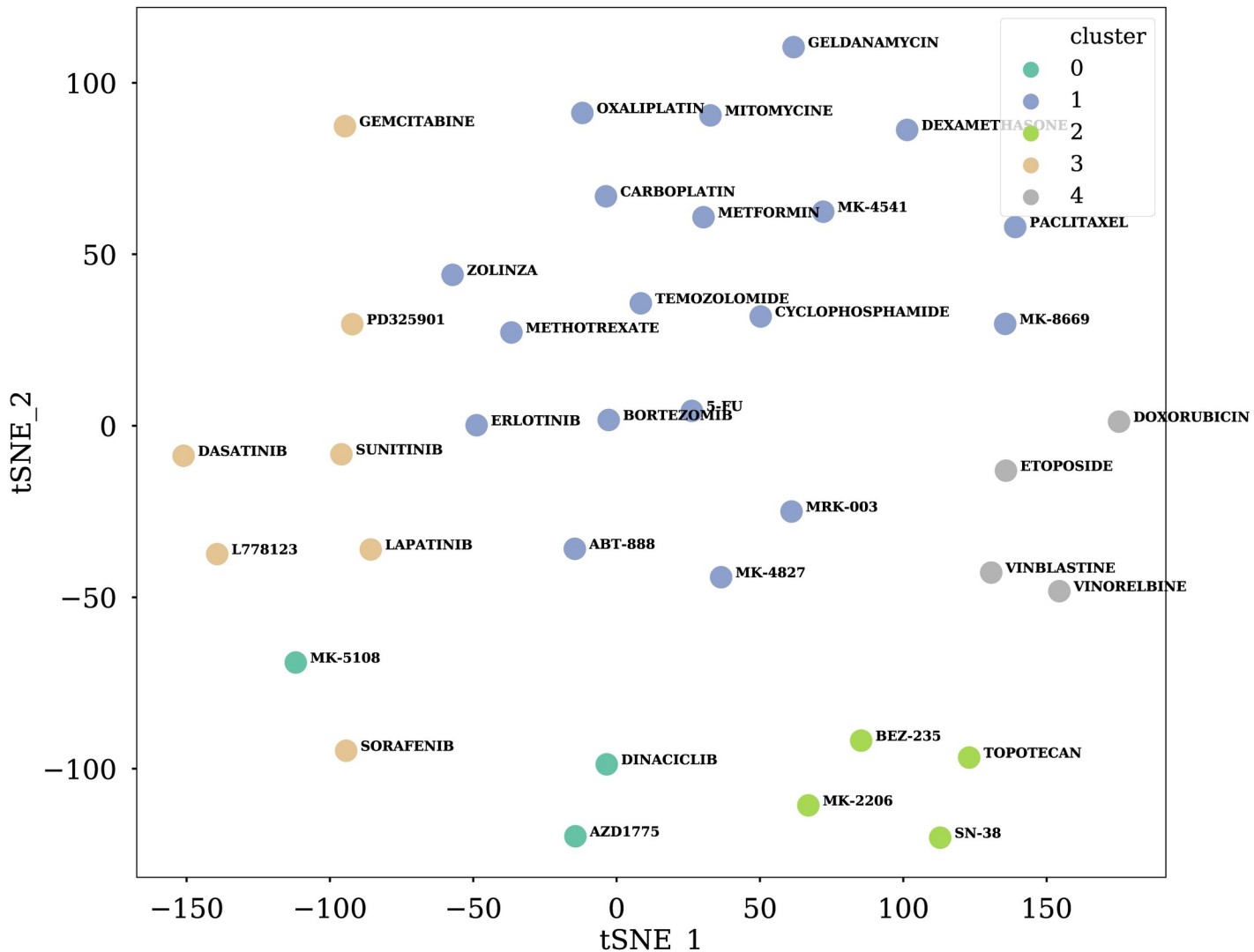

**Fig 6. Visualization of drug features with t-SNE analysis.** High dimensional ECFP representations are projected into 2D space with the first two t-SNE components. Different colors indicate assigned clusters of each drug by affinity propagation method.

exists in most drug pairs that are synergistic on the CAOV3 cancer cell line. It is consistent with the results of SA-GSEA in which the target combination of TOP2A, TOP2B, and TUBB shows high feature importance. Two samples show synergistic effects on OV90, another

**Table 5. Ablation study of TranSynergy models.**

| Drug features | Cell line features | MSE | Spearman Correlation | Pearson Correlation |
|---|---|---|---|---|
| Network propagated drug target profile | Gene dependency | 232 ± 21 | 0.725±0.016 | **0.746±0.017** |
| | Gene expression | 241 ± 19 | 0.721±0.021 | 0.741±0.025 |
| | Gene dependencies + Gene expression | 231 ± 21 | **0.730±0.016** | **0.746±0.018** |
| | None | 390 ± 11 | 0.505±0.007 | 0.400±0.012 |
| Observed drug target profile | Gene dependency | 305 ± 34 | 0.634±0.035 | 0.635±0.048 |
| | Gene expression | 307 ± 33 | 0.634±0.038 | 0.632±0.039 |
| | Gene dependencies + Gene expression | 300 ± 31 | 0.645±0.036 | 0.634±0.038 |

**Table 6. The most important oncogenic signatures revealed by SA-GSEA.**

| Drug combination (drug 1, drug 2) | Cell line | Targets with high SHAP values | | The most enriched gene set in the cell line |
|---|---|---|---|---|
| | | Drug 1 | Drug 2 | |
| BEZ-235, ERLOTINIB | T47D | Targets in PI3K/AKT/mTOR pathway | EGFR | oncogenic signature for RAF overexpressed cells |
| BEZ-235, MK-4827 | T47D | Targets in PI3K/AKT/mTOR pathway | PARP1/2 | oncogenic signature for RAF overexpressed cells |
| BEZ-235, L-778123 | T47D | Targets in PI3K/AKT/mTOR pathway | FPTase/ GGPTase | oncogenic signature for RAF overexpressed cells |
| BEZ-235, DINACICLIB | T47D | Targets in PI3K/AKT/mTOR pathway | CDKs | oncogenic signature for RAF overexpressed cells |
| ETOPOSID, MK-8669 | CAOV3 | TOP2A/TOP2B | mTOR | oncogenic signature for JAK2 knock-down cells |
| PACLITAXEL, DASATINIB | MSTO-211H | Tubulin/microtubule associated proteins | BCRABL, SRC, Ephrins and GFR | oncogenic signature for ESR1- cells |
| ABT-888, DASATINIB | MSTO-211H | PARP1/2 | BCRABL, SRC, Ephrins and GFR | oncogenic signature for SNF5 know-down cells |

ovarian cancer cells. For ETOPOSIDE and VINORELBINE combination, drugs also target TOP2A, TOP2B, and TUBB. For PACLITAXEL and VINORELBINE, both drugs target tubulin or microtubule-associated proteins. DEXAMETHASONE exists in synergistic drug combinations in three cell lines, OCUBM, SKMES1, and KPL1.

## Discussion

In this study, we have presented a novel deep learning model TranSynergy for the synergy score prediction and mechanism deconvolution of drug combination cancer therapy. We have demonstrated that the network propagated drug target profile, which indicated both drug-target interaction and drug effect on non-targeted proteins, was crucial for the comprehensive representation of drug features. The use of drug target information as drug features in the model might cause limitation when novel drugs, whose molecular targets are unknown, are involved in the drug pair. However, this limitation can be partially overcome by using recently developed methods for the accurate prediction of drug targets, e.g. [48]. In addition, drugs used in the drug combinations are mainly existing drugs whose targets are largely known, at least, it is true in the dataset used in this study. Because the drug target provides crucial information on the mechanism of drug action, it not only improves the accuracy of the machine learning model for the prediction of drug combination synergy but also facilitates the model

**Table 7. Examples of predicted synergistic novel drug pairs.**

| Drug combination | Cell line | Predicted Synergy score | Z-score |
|---|---|---|---|
| DEXAMETHASONE and ETOPOSIDE | CAOV3 | 94.520215 | 3.07259459 |
| ETOPOSIDE and METFORMIN | CAOV3 | 70.556332 | 2.21009253 |
| ETOPOSIDE and SN-38 | CAOV3 | 69.380475 | 2.16777138 |
| DEXAMETHASONE and VINBLASTINE | OCUBM | 38.6095235 | 1.94930657 |
| DEXAMETHASONE and PACLITAXEL | SKMES1 | 37.9697725 | 1.91635083 |
| ETOPOSIDE and VINORELBINE | OV90 | 33.844834 | 1.89400207 |
| PACLITAXEL and VINORELBINE | OV90 | 32.5451685 | 1.81387084 |
| ETOPOSIDE and VINBLASTINE | CAOV3 | 56.799048 | 1.71494381 |
| DEXAMETHASONE and VINBLASTINE | KPL1 | 37.8084735 | 1.6825975 |
| ETOPOSIDE and PACLITAXEL | CAOV3 | 53.1252895 | 1.58271882 |

interpretation. We also show that gene essentiality in the cancer cells is a more desirable cell line representation than raw gene expression profile. Due to the limited data size, we only selected the minimum number of genes for the representation of drugs and cell lines, including only drug targets and annotated cancer-related genes. When too many input features are used, the model could suffer overfitting problems during the training stage due to the curse of dimensionality [49,50]. The performance can be further improved when more data are available or by utilizing unlabeled data sets so that more genes can be included in the representations. Furthermore, we demonstrated that the performance in cold-start problems, including leave cell out and drug out scenarios, is relatively lower than the leave drug combination out scenario. We believe that the relatively lower performance in the cold-start setting could be improved by applying pretraining strategies.

With more and more high throughput drug combination screening datasets becoming available, we have to figure out the inconsistency problem in both the experimental and quantification methods for the determination of drug combination effects. Firstly, the combinational spaces for the drug doses used to generate the drug dose matrix diversify in different studies. Secondly, several distinguished methods were proposed to calculate the expected drug combination effect from experimental data, such as combination index (CI)-isobologram equation [51–53], Bliss independence (BI) method [54–56], and Loewe Additivity (LA) model [57,58]. The calculated drug synergy scores are not the same when different quantification methods are utilized. To further improve the data quality, it is necessary to develop new methods to harmonize different data sets.

Deep learning-based computational models have made promising breakthroughs in many biomedical areas. Interpretation of deep learning models becomes critical to overcoming the skepticism of it being a black-box [59]. Recently, many explainable AI methods have been proposed, such as input perturbation methods [60,61], backpropagation based methods [62], and the calculation of SHAP values [33]. Attention-based approaches have been proposed to interpret the models using attention mechanisms. However, previous work has shown that the attention weights learned from self-attentions may not provide a meaningful explanation for the final prediction [63]. As a result, we carefully design the input features so that each feature dimension is corresponding to a gene and is easily interpretable. Through examining the SHAP values of gene-wise input feature, we extract the information regarding the effect of drug-target interactions and gene-gene interactions on the cancer cell response. Considering that evidence has shown synergistic effects could be attributed to either pathway cross-talking [64], nonoverlapping pathways [28], or same pathways, the explanation of drug combination synergy score learned with our model could come from either of instances. More conclusive statements for the underlying mechanisms might require additional experimental evidence, which is out of our work's scope. However, we suggest that this potentially provides a method to generate testable hypothesis for studying the underlying mechanism of the drug combinations therapy.

Drug combinations can be a more efficient therapeutic strategy for cancer by targeting multiple proteins to defer the rapid emergence of drug resistance. The exploration of effective and synergistic drug combinations is hindered by the costly and time-consuming experimental preclinical investigation. Computational methods can be a cheaper and faster alternative approach to facilitate the development of drug combination therapy for cancer patients [65]. Nowadays, more emphasis is put on personalized medicine, which requires the consideration of the heterogeneity of each patient's cancer types and genomics information to find more efficient therapy. Given a large amount of data for patients' genome information, the development of an accurate and interpretable computational model is critical for the realization of

personalized medicine. Mechanism-driven machine learning as demonstrated in this study is a promising direction to address challenges in precision medicine of combination therapy.

## Methods

### Drug combination synergy score dataset

The large-scale drug combination screening dataset was initially published by Merck [20] and preprocessed to calculate the synergy scores [24]. The screening test was performed with 38 drugs and 39 cancer cell lines. In total, 583 pairs of drug combinations were investigated and 23062 data points were collected. Among them, two drugs lack target information, and the gene dependency and gene expression data of four cells are not available. These drugs and cell lines were excluded from our data set. We finally selected 36 drugs that targeted at least one protein and 35 cell lines. The final dataset has 18553 data points and 523 pairs of drug combinations.

### Drug representation

The observed drug target profile was collected from two datasets, Drugbank and ChEMBL, for the 36 drugs [36,37]. The observed drug target matrix is a $36^{*}2401$ binary matrix that indicates whether a drug targets a protein. The observed drug target profile was processed with the RWR algorithm to obtain a novel network propagated drug targets profile. We solved the RWR problem with the Fast RWR methods [66]. Following is the formal equation:

$$r = \alpha W r + (1 - \alpha)e,$$

$\alpha$ is a hyperparameter equal to 1—restart rate. To avoid information leaking and overfitting, we performed nested cross-validation for hyperparameter tuning and model selection. In each iteration, the data were split into training/validation/test sets. After the model was fitted using the training data set, optimal hyperparameter $\alpha$ in RWR was selected by the grid search over the validation set. The grid search determined the optimal $\alpha$ = 0.5 (S6 Table) over the validation set. W is the transition matrix denoting the transition probability between nodes. We used the protein-protein interaction network matrix from STRING as the transition probability matrix [67]. The edge weight between nodes is the protein-protein association confidence score. $e$ is the seed vector, a drug target binary vector for each drug in this Eq 1, denotes that the drug is targeting the protein. $r$ is the final probability distribution of each node in the network. Intuitively, $r$[i] denotes the effect of the drug on each protein. Drug's atoms and edges information were extracted with rdkit from the drug's SMILES string. We then used the multi-hot encoder method to represent each node and edge. The final representation includes 62 atoms and 6 bonds information.

### Cell line representation

Gene expression profiles were downloaded from Harmonizome [68] and were initially collected in Broad Institute Cancer Cell Line Encyclopedia (CCLE) and Genomics of Drug Sensitivity in Cancer (GDSC) [69–71]. Gene dependencies profile dataset is a combined dataset from the Broad Institute Project Achilles [72–74] and Sanger CRISPR data from Wellcome Trust Sanger Institute and Broad Institute[75,76]. The dataset was downloaded from the Dep-Map portal [38]. The linear regression imputation method was used to fill in the missing value in the dataset with the MICE package [77].

## Model architecture

**Transformer.** The transformer has been widely used and shown promising performance in many different applications, including natural language processing and image processing [29–31]. It includes two major components, encoder and decoder. The input for both encoder and decoder has three matrices, query, key and values. Both encoder and decoder contain many sublayers. Each sublayer is comprised of three stages, attention mechanism, add & norm stage, and feed forward stage. The add & norm stage contains a residual structure and a layer normalization structure. The attention mechanism is the module to encode the interaction of different features with the following equation:

$$Attention(Q, K, V) = softmax(Q * K^T / \sqrt{(d)}) * V$$

where $Q$, $K$, and $V$ are the query, key and, values matrices, d is the dimension of hidden vector representation and softmax is the activation function.

**Graph convolutional neural network.** We applied the graph convolutional network (GCN) module to extract drug neural fingerprint in our TranSynergyGNF model (S2 Fig). Each drug is parsed as a graph, consisting of nodes that are the chemical atoms, and edges that are the chemical bonds. The GCN was used to extract and integrate the information of nodes and edges in each node's neighborhood. The output hidden representation can be interpreted as a representation of chemical substructure within the multi-hop distances from each node. The output of each GCN layer can be calculated with the following equation:

$$GCN(H^l, A) = softmax(D^{-1}(A + I)H^l W^l),$$

where $A$ is the adjacency matrix for the graph structure, $I$ is an identity matrix, $D$ is the diagonal node degree matrix of $A+I$, $H$ is the hidden representation from the previous layer, $W$ is the learnable parameter matrix and softmax is the activation function. To construct the hidden representation of the whole drug, we concatenate the hidden representation for each node to a single vector.

## Model evaluation

Model evaluations were performed in three scenarios. a) In the leave drug combination out scenario, we applied the nested cross-validation method. The data was split into five folds such that the drug pair in one fold does not overlap with the drug pairs in other folds (Fig 3A). Because the drug combination of drug A with drug B and drug B with drug A should have the same drug combination synergy score, the size of training data was doubled by swapping the drug A and drug B. The model was iteratively trained/validated in four folds and then tested in the remaining one fold. b) We further tested our models in the most challenging leave cell out scenario. We used affinity propagation methods to group the cell lines into 4 clusters based on cell line features similarity (Fig 3B). Since 28 cell lines are in the biggest cluster, we used them as the training/validation dataset and held out the remaining cell lines in other clusters as the test dataset (Fig 5). In this way, the cell lines in the testing set are significantly different from those in the training/validation set. c) In the leave drug out scenario, we grouped the drugs into 5 clusters with the affinity propagation method based on their structural similarities (Figs 3C and 6). We chose drugs in one cluster and held out all drug pairs including these three drugs as the test set. The remaining drug pairs are the training/validation set. We performed three tests with keeping out the drugs in each of the three smallest clusters. In each test, the ratio of samples in the training and the testing data is around 4:1.

In all the above scenarios, the MSE was used as the training loss. We then investigated the Spearman correlation and Pearson correlation between predicted data and ground-truths. ROC-AUC and PR-AUC were also used to evaluate the model performance for the classification task. Drug pairs with synergy scores larger than 30 were classified as positive pairs and those with lower synergy scores were classified as negative pairs.

## 2.5 Shapley additive gene set enrichment analysis

We used the GradientExplainer and DeepExplainer in the SHAP package to calculate Shapley value that characterizes the contribution of each input feature to the final prediction [33]. We used k-means to summarize the total dataset as the background dataset. The final Shapley value of each input feature was the average value of 10 tests for each sample data. We then ranked genes based on the Shapley values for the gene-wise features of cell line representation and conducted gene set enrichment analysis to unveil the enriched gene sets with GSEA [78,79].

## Supporting information

**S1 Table. The hyperparameters of TranSynergy model with the best performance.**
(XLSX)

**S2 Table. Performance of TranSynergy integrated with Chemical Information (TranSynergyCI).**
(XLSX)

**S3 Table. Performance of TranSynergy integrated with Graphical Neural Fingerprint (TranSynergyGNF).**
(XLSX)

**S4 Table. Sample amounts in each cell line.**
(XLSX)

**S5 Table. Predicted drug pairs synergy score and z-scores.**
(XLSX)

**S6 Table. Grid search results of the restart rates $\alpha$, a hyperparameter in random walk with restart for network propagated drug target features inference, over the validation set.**
(XLSX)

**S1 Fig. The architecture of TranSynergy integrated with Chemical Information (TranSynergyCI).** Extra drug features, including ECFP, physico-chemical and toxicophore features were concatenated with the output of the Transformer and input into the last fully connected component. These chemical features are the same as those used in deepSynergy.
(PDF)

**S2 Fig. The architecture of TranSynergy integrated with Graphical Neural Fingerprint (TranSynergyGNF).** Extra drug features, graphical neural fingerprint for drug A and drug B extracted with a GCN layer, were concatenated with the output of the Transformer and input into the last fully connected component.
(PDF)

**S3 Fig. Shapley Additive Gene Set Enrichment Analysis result of the prediction model for Etoposide and MK-8669 drug pair.** The left panel shows the SHAP values of drug targets,

while the right panel shows the SHAP values of 20 genes with the most significant impact. (PDF)

**S4 Fig. Shapley Additive Gene Set Enrichment Analysis result of the prediction model for Erlotinib and BEZ-235 drug pair.** The left panel shows the SHAP values of drug targets, while the right panel shows the SHAP values of 20 genes with the most significant impact. (PDF)

**S5 Fig. Shapley Additive Gene Set Enrichment Analysis result of the prediction model for MK-4827 and BEZ-235 drug pair.** The left panel shows the SHAP values of drug targets, while the right panel shows the SHAP values of 20 genes with the most significant impact. (PDF)

**S6 Fig. Shapley Additive Gene Set Enrichment Analysis result of the prediction model for L778123 and BEZ-235 drug pair.** The left panel shows the SHAP values of drug targets, while the right panel shows the SHAP values of 20 genes with the most significant impact. (PDF)

**S7 Fig. Shapley Additive Gene Set Enrichment Analysis result of the prediction model for Paclitaxel and Dasatinib drug pair.** The left panel shows the SHAP values of drug targets, while the right panel shows the SHAP values of 20 genes with the most significant impact. (PDF)

**S8 Fig. Shapley Additive Gene Set Enrichment Analysis result of the prediction model for BEZ-235 and Dinaciclib drug pair.** The left panel shows the SHAP values of drug targets, while the right panel shows the SHAP values of 20 genes with the most significant impact. (PDF)

**S9 Fig. Shapley Additive Gene Set Enrichment Analysis result of the prediction model for Dasatinib and ABT-888 drug pair.** The left panel shows the SHAP values of drug targets, while the right panel shows the SHAP values of 20 genes with the most significant impact. (PDF)

## Author Contributions

**Conceptualization:** Qiao Liu, Lei Xie.

**Data curation:** Qiao Liu, Lei Xie.

**Formal analysis:** Qiao Liu, Lei Xie.

**Funding acquisition:** Lei Xie.

**Investigation:** Qiao Liu, Lei Xie.

**Methodology:** Qiao Liu, Lei Xie.

**Project administration:** Lei Xie.

**Resources:** Lei Xie.

**Software:** Qiao Liu, Lei Xie.

**Supervision:** Lei Xie.

**Validation:** Qiao Liu, Lei Xie.

**Visualization:** Qiao Liu, Lei Xie.

**Writing – original draft:** Qiao Liu, Lei Xie.

**Writing – review & editing:** Qiao Liu, Lei Xie.

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
