## [Decision Letter · Decision Letter 0]

22 Sep 2020

Dear Dr. Xie,

Thank you very much for submitting your manuscript "TranSynergy: Mechanism-Driven Interpretable Deep Neural Network for the Synergistic Prediction and Pathway Deconvolution of Drug Combinations" for consideration at PLOS Computational Biology.

As with all papers reviewed by the journal, your manuscript was reviewed by members of the editorial board and by several independent reviewers. In light of the reviews (below this email), we would like to invite the resubmission of a significantly-revised version that takes into account the reviewers' comments.

We cannot make any decision about publication until we have seen the revised manuscript and your response to the reviewers' comments. Your revised manuscript is also likely to be sent to reviewers for further evaluation.

Sincerely,

Avner Schlessinger

Associate Editor

PLOS Computational Biology

Feilim Mac Gabhann

Editor-in-Chief

PLOS Computational Biology

Reviewer's Responses to Questions

**Comments to the Authors:**

Reviewer #1: In this paper, the authors proposed a novel knowledge-enabled and self-attention boosted deep learning model to investigate synergistic drug combinations. They used random walk with restart (RWR) algorithm on a PPI network (drug features) and PPI derived NetExpress scores (gene dependency) as input to their model. They also perform gene set enrichment analysis to reveal novel biological pathways associated with the drug combinations using genes ranked based on the shapely additive value. TranSynergy has three components in its architecture, feature reduction, self-attention transformer that models gene-gene interaction and a fully connected neural network for final drug combination synergy score prediction. They compared their algorithm to another one named “Deepsynergy” and used the same train test split to attain complete compatibility. Their algorithm shows significant improvement in performance over DeepSynergy. The paper is well written and organized. However, the reviewer has the following concerns.

Major:

(1) Alpha is an important hyperparameter in RWR method. The authors show the impact of this parameter on the prediction results in Table S3 and use alpha = 0.5 in the experiments. Is this an 'overfitting'? The authors used the best results without tuning this hyperparameter to compare it to the other baseline methods. A detailed description is needed.

(2) The authors used RWR on a PPI network to represent the drug features instead of chemical information. It will be interesting to see the prediction results by integrating both drug features and chemical information in the TranSynergy model.

(3) It is also better to show the baseline performance in Figure 3, since the improvement from TranSynergy can be cancer specific.

Minor:

(1) It is better to show that whether the improvement by TranSynergy in Table 1 and Table 2 are statistical significance.

Reviewer #2: Summary

Authors Liu and Xie create a machine learning algorithm, TransSynergy, that aims to identify synergistic drug combinations by using information derived from drug modulation of protein-protein interaction networks. The authors claim to outperform drug-drug synergy predictions from a model created by called DeepSynergy and demonstrated how their Shapley additive gene set enrichment analysis could be used to deconvolute the transynergy black box and identify pathways associated with synergistic drug combinations.

Review:

The approach the authors describe is quite novel. Encoding drugs not as chemical descriptors but as a weighted network of their protein protein interactions is very innovative; considering most other approaches that are used in this field utilize descriptors such as ECFP4 and etc. As the authors correctly point out, it may be difficult to explain causal relationships and is not indicative of drug action at a cellular level.

In terms of performance, the authors claim that TranSynergy along with certain features can outperform deepsynergy. However there are some issues, first the main text does not include the number of cross validation steps the authors performed. Second, deepsynergy includes several other benchmarks other than correlation coefficients, such as mean squared error, ROC AUC, precision, recall, etc. that the authors do not include in this analysis. Because of this, their approach is not an apples-to-apples comparison, as the authors claim. This issue is persistent through to figure 3 and table 4, which only includes correlation coefficients and does not include other metrics of classifier performance.

Next the authors attempt to describe the cell line specific performance of their approach in figure 3. It is unclear in the main text the approach they used to predict cell-line specific performance. Did they train transsynergy on all other cell line data and test on a new cell line, or did they take 80% of all cell line data, and they are showing the performance on the hold out set? If it’s the latter, then confidence intervals should be shown. It would also be important to show the number of samples available for each cell line. Perhaps the low performance on prostate tissues may be due to low training data available for prostate cells, or the model is overfit to other cell lines.

Finally the authors show that using their variant of GSEA can identify pathways involved in the PPI of drugs. What would be helpful here would be a comparison of their shapely method and typical GSEA results. And observing the differences between the top pathways associated with either.

Other general comments:

While the author’s approach for representing drugs as networks of their PPI is interesting, there might be important chemical properties that the authors might be missing by not incorporating chemical features.

The deepsynergy paper uses more robust cross validation strategies than presented in this paper. Why wasn’t those strategies used in this work?

There are numerous grammatical errors present throughout the paper.

Reviewer #3: This manuscript introduced the development of a deep learning-based method for cell line-specific synergy prediction for drug combinations. It also introduced the application of Shapley Additive Gene Set Enrichment Analysis that can deconvolute the prediction to determine the synergy-associated biological pathways. The major contributions are as follows:

A. Compared to the earlier work of DeepSynergy, this study has adopted unique input features for both drugs and cell lines. Specifically, drugs are not represented as chemical features of drugs themselves but as gene targets (with gene-gene network propagation) of drugs, which helps the post-hoc interpretability of the model. Cell lines are not represented as gene expressions but experimentally determined or computationally predicted gene dependencies of cell lines, which again helps model interpretability.

B. The deep learning model uses self attention-empowered transformer. A post-hoc interpretability analysis further reveals gene sets associated with predicted synergy.

Numerical experiments show that the resulting TranSynergy made improvement compared to DeepSynergy (Pearson's r increased from 0.73 to 0.77 and Spearman's rho increased from 0.69 to 0.75). Ablation studies show that both the cell line features and the self attention modules contributed to the improvement. TransSynergy was split-analyzed based on tissues, interpreted with the gene enrichment analysis, and prospectively applied to predict for 3650 novel samples.

Major comments:

1> The use of targets to featurize drugs, although helping model interpretability, limits the applicability of the model. Unlike DeepSynergy that only assumes chemical information of the drugs themselves, TranSynergy demands the drug-target association whereas such data is not always available to drugs or compounds. Meanwhile, just using the new compound features with neither new cell line features nor self attention (Ablation Study) led to similar performance as DeepSynergy.

2> Please include more details about what DeepSynergy model was used and how it was derived. For fair comparison, DeepSynergy is supposed to be re-trained using the same training set (the original study used cross validation instead) and the training procedures ought to be provided as well.

3> Is it possible to examine the generalizability of the model under various scenarios of the test set, including new drugs (one or both in the pair), new targets (one or both in the pair), and new cell lines not seen in the training set. Performances on a new dataset (for instance, part of DrugComb) with such scenario splits would be useful. If not, performances on the existing test set with such split could help as well.

4> The performance analyses on different tissues are very interesting and worthwhile further interpretation. To help understand model behaviour, what could have made predictions be high mean and high variance for ovarian cell lines whereas low(er) mean and low variance for prostate cell lines?

5> Interpretation of the model predictions.

a. Self attention was mentioned for encoding drug-target interactions on Page 9 but for representing gene-gene interactions on Page 11. Is there a conflict or the two statements are equal because drugs are represented as target genes (and others in biological networks)?

b. Moreover, why not use learned self-attentions to interpret the model behavior but use the post-hoc Shapley Additive Gene Set Enrichment Analysis? Or maybe self-attentions were indeed used (although not documented in the enrichment analysis part of the methods)?

c. The interpretability results emphasize "pathways" associated with the synergy. But the results seem more about genes. It is not clear that pathway information was used for associated pathways, after finding those genes. Also, conceptually, could any explanation be introduced about what pathways are associated with synergy? The same pathway for both drugs? Two overlapping or cross-talking or nonoverlapping pathways covering the same disease module (consistent with the cell line)? Or something else?

Minor comments:

6> Please include more details about the 3650 novel samples in the section of methods.

7> With the current readme of the github repo, it is hard to figure out the structure of many source code files laid at the same top level. In addition, to help achieve more impact of the work, please consider including in readme how to train the model with the codes and which data files are what.

8> There were light grammar issues.

**Have all data underlying the figures and results presented in the manuscript been provided?**

Reviewer #1: Yes

Reviewer #2: Yes

Reviewer #3: Yes

PLOS authors have the option to publish the peer review history of their article (what does this mean?). If published, this will include your full peer review and any attached files.

Reviewer #1: No

Reviewer #2: No

Reviewer #3: No
---

## [Decision Letter · Decision Letter 1]

21 Dec 2020

Dear Dr. Xie,

We are pleased to inform you that your manuscript 'TranSynergy: Mechanism-Driven Interpretable Deep Neural Network for the Synergistic Prediction and Pathway Deconvolution of Drug Combinations' has been provisionally accepted for publication in PLOS Computational Biology.

Best regards,

Avner Schlessinger

Associate Editor

PLOS Computational Biology

Feilim Mac Gabhann

Editor-in-Chief

PLOS Computational Biology

Reviewer's Responses to Questions

**Comments to the Authors:**

Reviewer #1: The authors have addressed most of the concerns I had with the previous submission. I don't have any further comments

Reviewer #2: Authors sufficiently addressed my comments

Reviewer #3: The reviewer would like to thank the authors for the thorough revision. The additional experiments, analysis, clarification, discussion, and GitHub repo restructuring have adequately addressed the earlier comments.

**Have all data underlying the figures and results presented in the manuscript been provided?**

Reviewer #1: Yes

Reviewer #2: Yes

Reviewer #3: Yes

PLOS authors have the option to publish the peer review history of their article (what does this mean?). If published, this will include your full peer review and any attached files.

Reviewer #1: No

Reviewer #2: No

Reviewer #3: No

---

## [Editor Report · Acceptance letter]

2 Feb 2021

PCOMPBIOL-D-20-01357R1 

TranSynergy: Mechanism-Driven Interpretable Deep Neural Network for the Synergistic Prediction and Pathway Deconvolution of Drug Combinations

Dear Dr Xie,

I am pleased to inform you that your manuscript has been formally accepted for publication in PLOS Computational Biology. Your manuscript is now with our production department and you will be notified of the publication date in due course.

With kind regards,

Alice Ellingham
